# Health-Related Quality of Life (HRQoL) in Neuroendocrine Tumors: A Systematic Review

**DOI:** 10.3390/cancers14061428

**Published:** 2022-03-10

**Authors:** Rohit Gosain, Medhavi Gupta, Arya Mariam Roy, Jonathan Strosberg, Kathryn M. Glaser, Renuka Iyer

**Affiliations:** 1University of Pittsburgh Medical Center (UPMC) Hillman Cancer Center, UPMC Chautauqua Hospital, Jamestown, NY 14701, USA; gosainr@upmc.edu; 2Program in Women’s Oncology, Women and Infants Hospital and Warren Alpert Medical School of Brown University, Providence, RI 02912, USA; medhavi_gupta@brown.edu; 3Department of Medicine, Roswell Park Comprehensive Cancer Center, Buffalo, NY 14263, USA; arya.roy@roswellpark.org; 4Department of Gastro Intestinal Oncology, Moffitt Cancer Center, 12902 Magnolia Dr., Tampa, FL 33612, USA; jonathan.strosberg@moffitt.org; 5Department of Cancer Prevention & Control, Roswell Park Comprehensive Cancer Center, Buffalo, NY 14263, USA; kathryn.glaser@roswellpark.org

**Keywords:** health-related quality of life, patient-reported outcomes, neuroendocrine tumor, PRRT, octreotide, everolimus, lanreotide

## Abstract

**Simple Summary:**

Neuroendocrine tumors (NETs) are a group of heterogenous neoplasms arising from the diffuse neuroendocrine system. Several therapies have been added to the treatment landscape that have improved long-term outcomes. Despite therapeutic advancements, the symptom burden of the disease remains high, impacting health-related quality of life (HRQoL). In this study, we reviewed the impact of different treatment modalities on HRQoL in NET patients. Through a thorough literature review, 61 out of 2375 publications met the inclusion criteria. All randomized phase III trials leading to drug approvals showed a lack of deterioration of HRQoL, and one showed improved QoL. Capturing and understanding patient-reported outcome data is of vital importance for both patients and physicians to make treatment-related decisions.

**Abstract:**

Therapeutic advancements in neuroendocrine tumors (NETs) have improved survival outcomes. This study aims to review the impact of the current therapeutics on health-related quality of life (HRQoL) in NET patients. A literature review was performed utilizing PubMed, The Cochrane Library, and EMBASE, using the keywords “Carcinoid”, “Neuroendocrine tumor”, “NET”, “Quality of life”, “Chemotherapy”, “Chemoembolization”, “Radiofrequency ablation”, “Peptide receptor radionucleotide therapy”, “PRRT”, “Surgery”, “Everolimus”, “Octreotide”, “Lanreotide”, “Sunitinib”, and “Somatostatin analog”. Letters, editorials, narrative reviews, case reports, and studies not in English were excluded. Out of 2375 publications, 61 studies met our inclusion criteria. The commonly used instruments were EORTC QLQ-C30, FACT G, and EORTC- QLQ GI.NET-21. HRQoL was assessed in all pivotal trials that led to approvals of systemic therapies. All systemic therapies showed no worsening in HRQoL. The NETTER-1 study was the only study to show a statistically significant improvement in HRQoL in several domains. The trial examining sunitinib versus placebo in pancreatic NETs showed no change in QoL, except for worsening of diarrhea. In addition to clinical outcomes, patient-reported outcomes are a key element in making appropriate treatment decisions. HRQoL data should be readily provided to patients to assist in shared decision-making.

## 1. Introduction

Neuroendocrine tumors (NETs) are a group of heterogeneous malignancies that originate from neuroendocrine cells. Most well-differentiated NETs have an indolent course [1,2]. Multiple new treatments have been approved in recent years, primarily based on the evidence of inhibition of tumor growth and delay of disease progression [3,4,5,6,7,8]. Nevertheless, the symptomatic burden of disease remains high. A recent global survey of patients with NETs, conducted in more than 12 countries, demonstrated the considerable impact of NETs on symptoms, work, activities of daily living, and healthcare resource use [9]. NETs are associated with a wide range of symptoms from excess hormone production to tumor burden, which affect quality of life. The most commonly reported symptoms are fatigue, diarrhea, and flushing [1]. Clinical trials for new treatment options generally focus on disease progression and toxicity rather than quality of life. Moreover, treatment decisions are often based on patient comorbidities and the toxicity profile of the drugs, rather than taking patient preference or symptoms into consideration. Decision-making aimed only at improving overall survival with newer drugs may not achieve a desired symptom control goal or quality of life.

Patient-reported outcomes should play a pivotal role in selecting treatment options in NET, as symptoms can be debilitating. In recent years, health-related quality of life (HRQoL) measurements have gained increasing relevance in clinical trials. There are many available validated tools for evaluating HRQoL, but no clear consensus on the optimal tool for use in trials or clinical practice [10]. Given increasing awareness of the importance of HRQoL in the assessment of cancer treatment, we sought to perform a systematic review of studies measuring patient-reported QoL outcomes in NET patient studies.

## 2. Materials and Methods

We performed a thorough review of the published literature on health-related quality of life (HRQoL) in the NET population. Our search included the following databases: PubMed, The Cochrane Library, and EMBASE. The systematic research was performed in May 2019. The search combinations used included the Boolean expressions, “AND” and “OR” in combination with the following MeSH and free text terms: “Carcinoid”, “Neuroendocrine tumor”, “NET”, “Quality of life”, “Chemotherapy”, “Chemoembolization”, “Radiofrequency ablation”, “Peptide receptor radionucleotide therapy”, “PRRT”, “Surgery”, “Everolimus”, “Octreotide”, “Lanreotide”, “Sunitinib”, and “Somatostatin analog”. The same search strategy was employed in all 3 databases. Each of the eligible studies were reviewed, and data were extracted independently by two investigators to ensure consistency. Papers addressing HRQoL in NET patients were included in our analysis. Publications that were letters, editorials, narrative reviews, case reports, and studies not in English were excluded. The study is registered in the research registry for systematic analysis (unique identifying number-reviewregistry1315). The study selection process is described in Figure 1.

## 3. Results

### 3.1. HRQoL Study Selection & Characteristics

Our literature search revealed a total of 61 studies, which assessed the wellness of patient-reported quality of life (QoL) in NET patients using validated questionnaires. Of these 61 studies, nine were randomized controlled trials (RCTs), 16 were phase II clinical trials, 11 were prospective studies and the remaining 25 were observational studies. The RCTs and phase II clinical trials are listed in Table 1 [4,11,12,13,14,15,16,17,18,19,20,21,22,23,24,25,26,27,28,29,30,31,32,33,34]. The prospective studies and observational studies are listed in Appendix A [35,36,37,38,39,40,41,42,43,44,45,46,47,48,49,50,51,52,53,54,55,56,57,58,59,60,61,62,63,64,65,66,67,68,69,70,71]. In this literature review, we found two review articles focusing on HRQoL in NET patients [72,73]. The first review, by the Spanish NET Group, examined key reported outcomes in HRQoL studies in gastroenteropancreatic (GEP) NET patients. It evaluated different quality assessment tools used in various HRQoL trials and identified appropriate tools to assess HRQoL changes [72]. The second review by Martini et al. evaluated HRQoL studies in GEP NET patients from a methodological standpoint. Of the 48 eligible studies they examined, a range of methodological shortcomings was identified. They concluded that transferring HRQoL into practice is limited not just by the sparsity of studies but also due to the quality of HRQoL data processing [73].

All studies evaluated the quality of life in NET patients using validated QoL questionnaires. The commonly used questionnaires were: European Organization for Research and Treatment of Cancer Quality of Life Core Questionnaire (EORTC QLQ-C30), NET-specific QoL questionnaire (QLQ-GI NET-21), the Functional Assessment of Cancer Therapy-General (FACT-G), and the 12-Item Short Form Health Survey (SF-36).

### 3.2. QoL Instruments Used

The majority of the studies utilized EORTC QLQ-C30 (41/61 trials), an HRQoL questionnaire that is widely used in oncology trials [74]. The QLQ-C30 questionnaire is a multidimensional tool that has been translated and validated in over 100 languages and used in over 3000 studies worldwide. This questionnaire assesses the quality of life of patients on clinical trials using a series of 30 questions. These questions assess patient well-being based on five functional scales (physical, role, emotional, cognitive, social functioning), and several single- and multi-item symptom subscales (fatigue, nausea/vomiting, pain, dyspnea, insomnia, appetite loss, constipation, diarrhea, financial difficulties) [74]. Responses are translated into a 0–100-point scale to standardize the quality-of-life assessment. A QLQ-C30 score change of 5–10 denotes ‘a little’ change for better or worse on a particular scale (function or symptom), a score change of 10–20 denotes ‘moderate’ change and a change in score greater than 20 corresponds to ‘very much’ change from the baseline [75].

The EORTC has also developed a NET-specific QoL questionnaire, QLQ-GI.NET21. This questionnaire covers issues specific to NET patients that are not covered in the QLQ-C30 questionnaire, such as endocrine or specific gastrointestinal symptoms [64]. Nine of the studies used the QLQ-GI.NET21 questionnaire in conjunction with QLQ-C30. One study used the questionnaire EORTC QLQ Liver Metastases Colorectal (LMC21), in conjunction with QLQ-C30 [76].

The other commonly used questionnaire (n = 3 studies) was the Functional Assessment of Cancer Therapy-General (FACT-G), which comprises 27 items that assess patients’ physical, emotional, functional, and social well-being [77]. Other applied modules of FACT-G, FACT-Anemia, and FACT-hepatobiliary were also utilized by two studies. Another generic questionnaire used by two studies was the Functional Assessment of Chronic Illness Therapy (FACIT), and its modules FACIT-diarrhea and FACIT-fatigue.

Three studies used the Patient-Reported Outcome Measurement Information System 29-item Health Profile (PROMIS-29). This questionnaire assesses the quality of life of patients spanning seven important aspects–depression, anxiety, physical function, pain interference, fatigue, sleep disturbance, and the ability to participate in social roles and activities [78]. In one trial, the psychosocial adjustment to illness scale (PAIS) was used to assess patients’ psychosocial adjustment to their illness. The PAIS covers multiple domains including healthcare orientation, vocational environment, domestic environment, sexual relationships, extended family relationships, social environment, and psychological distress [79].

A total of 16 studies used either their self-made questionnaire, or other tools such as the 36-item Short Form Health Survey (SF-36), the 12-Item Short Form Health Survey (SF-12), the Euroqol-5 Dimension (EQ-5D), the French version of the Nottingham Health Profile (ISPN), and the General Health Questionnaire (GHQ-12). In general, these tools assess a patient’s quality of life from physical, social, and functional aspects.

### 3.3. HRQoL with Systemic Therapies

Currently approved therapies that are commonly utilized in NET patients include octreotide, lanreotide, everolimus, sunitinib, telotristat, and peptide receptor radionuclide therapy (PRRT) [4,7,13,65,80]. Pivotal randomized phase III trials using these agents evaluated patient quality-of-life while on active treatment or the control arm, utilizing validated HRQoL questionnaires. The study tools used, and results, are summarized in Table 2 and detailed below.

Two randomized phase III studies evaluated the role of somatostatin analogues (SSAs) in NET patients. The PROMID study randomized untreated, well-differentiated, metastatic mid-gut NET patients to receive either octreotide long-acting repeatable (LAR) versus placebo. QoL was assessed as a secondary endpoint using EORTC-QLQ C30 and described as a change from the baseline to after six months of follow-up. The HRQoL was assessed at random visits and at 3-month intervals until tumor progression. The symptoms of carcinoid syndrome were present in 39% of patients at the baseline. Symptomatic response was defined as a reduction of flushing to less than 1 flush/week, <4 bowel movements/week, and the absence of abdominal pain. There was no difference in the baseline mean EORTC-QLQ C30 score between the groups. At the 6-month follow-up mark, the symptomatic responses were higher in the octreotide LAR arm vs. placebo; however, none of these were statistically significant (flushing, 70% vs. 45%, *p* = 0.08; diarrhea, 33% vs. 19%, *p* = 0.56, abdominal pain 50% vs. 30%, *p* = 0.35) [13]. According to the post hoc analysis of the PROMID trial by Rinke et al., a statistically significantly longer time to deterioration (TTD) was observed in the octreotide LAR group than the placebo for fatigue (18.5 months vs. 6.8, *p* = 0.0006), pain, and insomnia. For diarrhea, there was an improvement in the change from the baseline to week 24 scores in the treatment group, and the score was worsened in the placebo group. Corresponding fatigue scores were stable in the treatment arm and worsened for placebo [82]. The phase III CLARINET study showed the efficacy of another SSA analogue, lanreotide, in comparison to placebo in advanced well/moderately differentiated, non-functioning, grade 1 (G1)/grade 2 (G2) gastroenteropancreatic NETs. The main adverse effect from the lanreotide was reportedly diarrhea, with 26% in the treatment arm having diarrhea when compared to 9% in the placebo arm. The second most common adverse effect was abdominal pain (14% vs. 2% in the treatment and placebo arm, respectively). HRQoL was a secondary endpoint and was assessed using EORTC QLQ-C30 and QLQ-GI.NET21 scores. There were no statistically significant differences in the change from the baseline to post-treatment QoL scores between the two treatment groups [4].

The phase III RADIANT-4 trial evaluated the efficacy of everolimus (mTOR inhibitor) in comparison to placebo in patients with advanced, well-differentiated, G1/G2 gastrointestinal and lung NETs. The Functional Assessment of Cancer Therapy-General (FACT-G) questionnaire was utilized to assess the HRQoL at baseline, every 8 weeks for the first 12 months, and every 12 weeks thereafter. The pre-specified secondary endpoint was the time to definitive deterioration of FACT-G score by at least seven or more points compared to the baseline. The median time to deterioration was similar between both the everolimus (11.3 months) and placebo arms (9.2 months), with no differences between the two groups (HR 0.81, 95% CI 0.55–1.21; *p* = 0.31) [17]. The most common side effects observed were stomatitis, followed by diarrhea and fatigue. In the treatment arm, 31% of patients had diarrhea with everolimus, compared to 16% observed in the placebo arm. In the everolimus group, grade 3 diarrhea was observed in 6% and grade 4 in 1% of patients, whereas no grade 4 diarrhea was seen in the placebo, and grade 3 was only 2%. Fatigue was also observed more frequently with everolimus when compared to placebo (31% versus 24%) [17,80].

Another targeted agent, sunitinib, a tyrosine kinase inhibitor, was approved in advanced, well-differentiated pancreatic NETs following a prospective trial demonstrating efficacy. In the randomized phase III trial of sunitinib versus placebo, the median progression free survival (PFS) on the sunitinib arm was 11.4 months, and the PFS in the placebo group was 5.5 months, with a hazard ratio (HR) of 0.42 (95% CI 026–0.66; *p* < 0.001). Functioning tumors were present in 51% of the sunitinib group and in 48% of the placebo group. The frequent adverse effects in the sunitinib group were diarrhea, nausea, vomiting, and fatigue. Diarrhea was seen in 50% of the treatment arm and 39% in the placebo arm. Grade 3 or 4 diarrhea was found in 5% of patients in the treatment arm and 2% in the placebo arm [6]. Quality of life was also a secondary endpoint of the study. EORTC QLQ-C30 scores were assessed at the baseline, day 1 of every cycle, and analyzed for the first 10 cycles. Baseline HRQoL scores were similar in the two arms. Over the first 10 cycles, no differences were noted in the global HRQoL, physical, emotional, cognitive, role, social functioning, or symptom scales, except for diarrhea. There was a clinical and statistically significant worsening of diarrhea in the sunitinib arm with a difference of 21.4 points between the two arms (*p* < 0.001). Even though the sunitinib arm had a statistically significant worsening of insomnia (*p* = 0.04), this was not clinically significant as the difference between the groups was 7.8 (between-group clinically significant difference is defined as >10 points) [6,15].

The phase III NETTER-1 trial randomized advanced, progressive, somatostatin-receptor positive, G1/G2 midgut NET patients to ^177^Lu-Dotatate PRRT plus octreotide LAR vs. octreotide LAR alone. PFS was clinically and statistically higher in the Lu-Dotatate group (65.2% vs. 10.8% at 20 months) with a hazard ratio of 0.18 (95% CI: 0.11, 0.29; *p* < 0.0001). The overall response rate was also higher in the Lu-Dotatate group (18% vs. 3% *p* < 0.001). The most common adverse effects in the ^177^Lu-Dotatate group were nausea (59% vs. 12%) and vomiting (47% vs. 10%), which was attributed to the concurrent amino acid infusion. Fatigue was also higher in the Lu-Dotatate group when compared to the control group (40% vs. 25% *p* = 0.03). Side effects that may affect the HRQoL, including diarrhea, abdominal pain and flushing, were slightly higher in the Lu-Dotatate group, but this was not statistically significant. The rate of grade 3 and grade 4 side effects was similar in both groups [7]. It was recently found that the median overall survival in Lu-Dotatate group was 48 months and was 36.3 months in the control arm, but this was not statistically significant (HR = 0.84, 95% CI: 0.60, 1.17: *p* = 0.30) [83]. HRQoL was measured using EORTC QLQ-C30 and GI NET-21 questionnaires every 12 weeks till disease progression. Time to QoL deterioration (TTD) was defined as the time from randomization to first deterioration >/10 points on a 100-point scale for that domain. The TTD was statistically longer in the PRRT arm for a number of domains including global health (HR 0.41, *p* < 0.001), role functioning (HR = 0.58, *p* = 0.03), physical functioning (HR 0.52, *p* = 0.15), disease-related worries (HR 0.57, *p* = 0.018), body image (HR = 0.43, *p* = 0.006), diarrhea 0.47 (*p* = 0.01), pain (HR 0.57, *p* = 0.025), and fatigue (HR 0.62, *p* = 0.03). Median TTD was statistically significant in the favor of PRRT for the global health domain (22.7-month difference) and physical functioning domain (13.7-month difference) [18].

The placebo-controlled TELESTAR study studied the role of telotristat ethyl (TE), a tryptophan hydroxylase inhibitor in patients with carcinoid syndrome and >/4 bowel movements (BM) per day despite taking SSAs. Patients were randomized to TE 250 mg three times a day (TID), TE 500 mg TID or placebo (1:1:1) during a double-blind treatment period (DBTP), and all patients received TE 500 mg TID in an open label extension (OLE) to week 48. This study used a unique parameter of durable response defined as a BM frequency reduction of >/30% from baseline for >/50% of the time [3]. An exploratory analysis evaluated the relationship between HRQoL and durable responders (DR)/non-durable responders (NDR). At the end of the 12-week DBTP, 48/135 patients were DR. In the OLE phase, 29/35 DR maintained the DR, and 71 of DBTP-NDRs became OLE_DRs. The analysis showed that DR was associated with better symptom control than NDR in both DBTP and OLE. Similarly, DR had QoL improvements in EORTC QLQ-C30 global health status, nausea, vomiting, pain, diarrhea, and EORTC QLQ-GI.NET21 gastrointestinal symptoms than NDR both over the DBTP and OLE. The EORTC QLQ-C30 diarrhea subscale scores were much improved in the telotristat group. The score improvements for TE 250 mg, 500 mg groups were 19.2 points, 21.6 points, and that of the placebo was only 8.5 points [3,81].

## 4. Discussion

Patient-centered outcomes have become an important focus of research, and few physicians have had formal training in this aspect of patient evaluation. This form of research assesses the benefits and harms of preventive, diagnostic, therapeutic, palliative, and other interventions for shared decision-making. This knowledge further helps in tailoring the treatments based on the meaningful endpoint of quality of life and highlights outcomes that are relevant to patients and caretakers which eventually lead to treatment satisfaction. Most clinical trials and retrospective studies focus on the survival benefits and toxicity profiles of the treatments rather than impact on quality of life. In a patient-centric treatment model, treatment discussions should also include data on HRQoL outcomes. Decisions informed by HRQoL can help to improve the patients’ physical and mental health and decrease the burden on health care by minimizing frequent hospital admissions and health care utilization [84].

HRQoL is a particularly important consideration for patients with NETs, who can develop both tumor- and hormone-related symptoms during their disease course [9]. The common symptoms affecting the HRQoL of NET patients are listed in Table 3. Patients with relatively slow-growing tumors may be prescribed local and systemic treatments for prolonged periods lasting a decade or more, and many hope to remain functional throughout their journey. Further, although the incidence of NET in patients 65 years or above is highest, it is also diagnosed in younger adults, whose health goals may be different [2].

HRQoL studies assessing locoregional modalities of surgery or interventional radiology (IR) procedures are limited by size, absence of a formal HRQoL analysis, and/or lack of prospective analysis. Therefore, a definitive impact of these procedures in the HRQoL of NET patients is largely unknown. There are multiple studies on the effect of surgeries and IR techniques for treating hepatic metastasis such as embolization, radiofrequency ablation, and cryoablation in patients with NET [25,26,44,46,55]. Very few surgical or IR studies have studied the impact of the procedures on the patient’s quality of life. Most of them lack a formal HRQoL analysis.

Validated tools and uniform definitions are key for assessing HRQoL and allowing comparison between studies of approved therapy options to aid treatment selection. There is a wide variety of tools and patient surveys for assessing HRQoL of cancer patients [10] in trials and in clinical practice. As summarized here, all the key phase 3 studies in NET patients used validated tools to assess HRQoL. Most of the studies used the EORTC QLQ-C30 questionnaire, which is a general oncology questionnaire. As fatigue and diarrhea are common in NET patients at the baseline, some trials used FACIT-fatigue and FACIT-D scoring while some studies used QLQ-GI.NET21, which is a NET-specific questionnaire. Cross-study comparisons will be easier with uniform adoption of HRQoL tools for future use, and facilitate treatment-choice discussions in clinical practice. One limitation, even with these validated tools, is reliance on recall when patients and caregivers fill out the HRQoL. A majority are done during office visits, and do not always have journals to track changes in the HRQoL events between visits. Besides journals to assist in recall, the availability of mobile phone applications, which can collect real-time data, may increase patient compliance and confidence in data capture from self-reporting [85].

While using a global score and a prior definition for improved QoL are important, there are also some limitations with looking only at overall scores. In the multiple practice-changing clinical trials leading to the approvals of octreotide LAR, lanreotide auto gel, everolimus, and sunitinib, no decline of QoL global scores was seen compared to the control arms, suggesting that therapy did not worsen the quality of life [4,6,13,17]. However, the trial examining sunitinib versus placebo in pancreatic NETs showed that, despite no change in QoL in all aspects, the diarrhea scores reflected that sunitinib worsened this quality measure. In addition, in a retrospective analysis of the PROMID trial, Rinke et al. observed a significantly longer time to deterioration in quality of life in the octreotide LAR group compared to the placebo in clinically relevant NET symptoms such as fatigue, diarrhea, pain, and insomnia, suggesting maintained or improved HRQoL with octreotide LAR [82]. Similarly, everolimus was also associated with more diarrhea and fatigue than the placebo group [80].

The NETTER-1 trial was the only study to have favorably impacted decline in HRQoL compared to control in key domains including global health, physical functioning, role functioning, fatigue, pain, and diarrhea, providing more granular insight on the impact of therapy on life. Therapy discussions should include these outcomes along with discussion of survival benefits [18].

Most trial publications focus on grade 3 and 4 events that result in dose reduction or discontinuation of therapy. However, chronic low-grade side effects such as fatigue can have a major impact on HRQoL. In the global health survey by Singh et al., the impact of the illness on employment was captured in those still working and 49% of patients reported taking days off work, 27% asked for accommodations, and 24% reduced work hours. In addition, among 82% of those who were unemployed, the reason for not working was disease-related symptoms [9]. Assessing financial impact is only partially included in most QoL assessments and is an important gap in knowledge for NET patients that future studies should aim to address. The EORTC QLQ-30 has clinical thresholds for physical functioning (<80), emotional functioning (<70), fatigue (>39), pain (>25), and role function (<90) scales based on literature [74]. These threshold scores provide valuable insight into clinically important problems that should be addressed and may impact not just treatment choices but also bring to light the need for timely referrals to supportive care services.

## 5. Conclusions

Health-related quality of life should be integrated into tailoring treatment selection in neuroendocrine cancer patients as patient-reported outcomes are a key element of shared decision-making and patient-centered care. In NET patients, the impact on QoL of all currently approved therapies compared to their control arms has been studied using well-validated tools, and should be provided to patients during their pretherapy counseling along with toxicity information. Toxicity assessment of specific treatments should not be a surrogate measure for quality of life of patients, and careful assessment of global QoL scores, as well as individual symptom scores, may provide key insights into the impact of the disease on quality of life and functioning. Real-life studies on the impact of therapy, where most of the care delivery occurs, and HRQoL studies during periods between active treatment need to be conducted to identify the true impact of therapy choices on patients living with NETs.

## Figures and Tables

**Figure 1 cancers-14-01428-f001:**
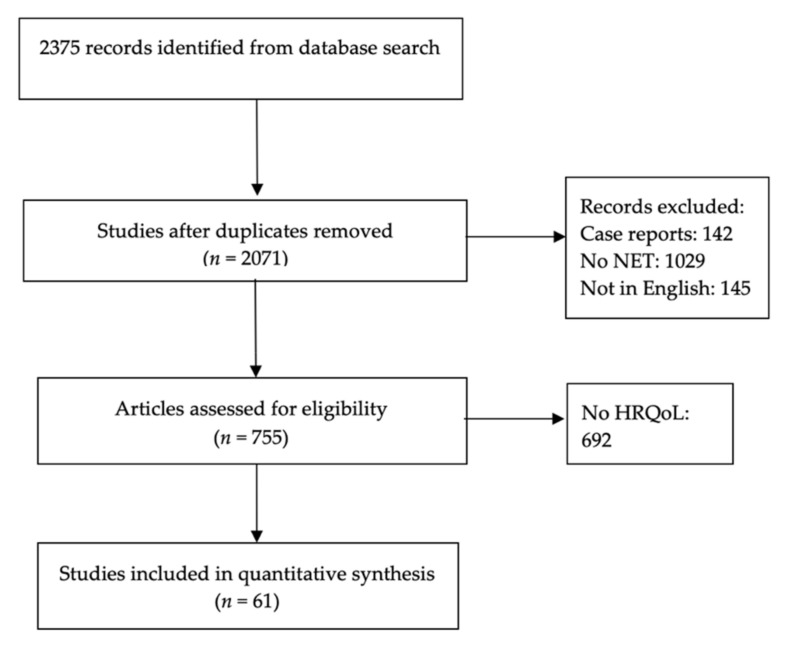
Flow chart describing systematic research search and study selection process.

**Table 1 cancers-14-01428-t001:** RCTs and phase II clinical trials in NET patients with reporting of HRQoL outcome measures.

Study	Type of Study	Sample Size	Treatment	Tumor Subtype	HRQoL Measure
Arnold et al., 2005 [11]	RCT	109	octreotide vs. octreotide + IFN	GI NETs	QLQ-C30
Bajetta et al., 2006 [12]	RCT	60	lanreotide autogel vs. lanreotide microparticle	GI NETs	QLQ-C30
Rinke et al., 2009 [13]	RCT	85	octreotide LAR vs. placebo	GI NETs	QLQ-C30
Caplin et al., 2014 [4]	RCT	204	lanreotide autogel vs. placebo	GI NETs	QLQ-C30, QLQ-GI. NET21
Meyer et al., 2014 [14]	RCT	86	capecitabine + streptozocin + cisplatin vs. capecitabine + streptozocin	GI NETs	QLQ-C30
Vinik et al., 2016 [15]	RCT	171	sunitinib vs. placebo	GI NETs	QLQ-C30
Vinik et al., 2016 [16]	RCT	115	lanreotide depot/autogel vs. placebo	GI NETs	QLQ-C30, QLQ-GI.NET-21
Pavel et al., 2017 [17]	RCT	302	everolimus vs. placebo	NETs	FACT-G
Strosberg et al., 2018 [18]	RCT	231	^177^Lu-DOTATATE vs. octreotide	GI NETs	QLQ-C30, QLQ-GI.NET-21
Wymenga et al., 1999 [19]	Phase II	55	lanreotide prolonged-release	NETs	QLQ-C30
Ruszniewski et al., 2004 [20]	Phase II	71	lanreotide prolonged-release	GI NETs	QLQ-C30
Zuetenhorst et al., 2004 [21]	Phase II	26	Interferon followed by meta- iodbenzylguanidin	NETs	QLQ-C30
Frilling et al., 2006 [22]	Phase II	18	^90^Y-DOTATOC; ^177^Lu-DOTATATE	NETs	SF-36
Kulke et al., 2008 [23]	Phase II	107	Sunitinib	NETs	EQ-5D, FACIT-Fatigue scale
Korse et al., 2009 [24]	Phase II	39	octreotide LAR	GI NETs	QLQ-C30
Bushnell et al., 2010 [25]	Phase II	90	^90^Y-DOTADOC	NETs	EQ-5D
Cwikla et al., 2010 [26]	Phase II	60	^90^Y-DOTATATE	GI NETs	QLQ-C30, QLQ-GI.NET-21
Bodei et al., 2011 [27]	Phase II	51	^177^Lu-DOTATATE	NETs	QLQ-C30
Claringbold et al., 2011 [28]	Phase II	33	^177^Lu-DOTATATE	NETs	QLQ-C30
Khan et al., 2011 [29]	Phase II	256	^177^Lu-DOTATATE	NETs	QLQ-C30
Kvols et al., 2012 [30]	Phase II	45	Pasireotide	GI NETs	FACIT-D
Martin-Richard et al., 2013 [31]	Phase II	30	lanreotide autogel	NETs	QLQ-C30
Delpassand et al., 2014 [32]	Phase II	37	^177^Lu-DOTATATE	GI NETs	QLQ-C30
Ducreux et al., 2014 [33]	Phase II	34	bevacizumab + 5FU/streptozocin	GI NETs	QLQ-C30
Mitry et al., 2014 [34]	Phase II	49	bevacizumab + capecitabine	GI NETs	QLQ-C30

Abbreviations: NET = neuroendocrine tumors, HRQoL = health-related quality of life, RCT = randomized controlled trial; GI = gastrointestinal, vs. = versus, IFN = interferon alpha; LAR = long-acting release, 5FU = 5-Flurouracil, SSA = Somatostatin Analogue, PRRT = peptide receptor radionuclide therapy, EORTC QLQ-30 = European Organization for Research and Treatment of Cancer Quality of Life Core Questionnaire, EORTC QLQ GI.NET21 = NET specific EORTC QoL questionnaire, FACT-G = Functional Assessment of Cancer Therapy-General, GHQ = general health questionnaire, SF-36 = 36-Item Short Form Health Survey, FACIT = The Functional Assessment of Chronic Illness Therapy.

**Table 2 cancers-14-01428-t002:** Health-related quality of life in phase III neuroendocrine tumor studies.

Clinical Trial	HRQoL Tool Used	Patient Population Studied	HRQoL in Comparison to Control Arm
Octreotide vs. Placebo (PROMID) [13]	EORTC QLQ-C30	GI and unknown primary NETs	At 6-month follow up mark, no statistically significant difference from baseline was observed between two arms.
Lanreotide vs. Placebo (CLARINET) [4]	EORTC QLQ-C30, EORTC QLQ-GI.NET21	GI, pancreatic NETs, and unknown primary	No statistically significant difference in the change from baseline to post treatment QoL scores between the two arms.
Everolimus vs. Placebo (RADIANT 4) [17,80]	FACT-G	GI and lung NETs	The median time to definitive deterioration in FACT-G score was similar between both arms with no significant difference between both arms.
Sunitinib vs. Placebo [6,15]	EORTC QLQ-C30	Pancreatic NETs	Over the first 10 cycles, no differences were observed in the global HRQoL, physical, emotional, cognitive, role, social functioning, or symptom scales except for diarrhea. Statistically significant worsening of diarrhea in the sunitinib arm with a difference of 21.4 points between the two arms was observed.
PRRT vs. Octreotide (NETTER 1) [7]	EORTC QLQ-C30, QLQ-GI.NET21	Midgut NETs	Time to QoL deterioration was statistically longer in the PRRT arm for multiple domains including global health, role functioning, physical functioning, disease-related worries, body image, diarrhea, pain and fatigue.
Telotristat vs. Placebo (TELESTAR) [3,81]	EORTC QLQ-C30 EORTC QLQ-GI.NET21	Carcinoid syndrome with diarrhea	Durable responders had QoL improvements in EORTC QLQ-C30 global health status, nausea and vomiting, pain, diarrhea, and EORTC QLQ-GI.NET21 gastrointestinal symptoms than non-durable responders both over the DBPT and OLE period *.

* Durable response defined as a BM frequency reduction of >/30% from baseline for >/50% of the time Abbreviations: NET = neuroendocrine tumor, GI = gastrointestinal, HRQoL = health-related quality of life, EORTC QLQ-30 = European Organization for Research and Treatment of Cancer Quality of Life Core Questionnaire, EORTC QLQ GI.NET21 = NET specific EORTC QoL questionnaire, FACT-G = Functional Assessment of Cancer Therapy-General, PRRT = peptide receptor radionuclide therapy, DBTP = double blind treatment period, OLE = open label extension.

**Table 3 cancers-14-01428-t003:** Common symptoms affecting HRQoL of NET patients.

Hormone-related symptoms	Diarrhea, flushing, fatigue, loss of appetite, dyspnea, palpitation, loss of weight
Tumor burden-related symptoms	Abdominal pain, abdominal distension, ascites, jaundice, compression of adjacent organs

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
