# Peer review of "Health-Related Quality of Life (HRQoL) in Neuroendocrine Tumors: A Systematic Review"

_cancers, 2022, doi:10.3390/cancers14061428_

Round 1

Reviewer 1 Report

The manuscript

Health Related Quality of Life (HRQoL) in Neuroendocrine Tumors: A Systematic Review”,

is a very nice and well written review article about highly important topic. The only point I am unhappy with is table one which is ranging over 6 pages! I do not think it is necessary to list the studies in a table like this. Please try to remove this table e.g. by creating a new table putting similar studies together or put more information in the text.

Otherwise please check the manuscript for minor errors in wording.

Author Response

Thank you very much for your valuable comments. We totally agree with the comments. We have made changes to table 1. We kept only randomized control trials and phase II clinical trials in table 1 and moved all the prospective, observational trials into another table in the supplementary file. Now, table 1 is only 1.5 pages.

Reviewer 2 Report

The authors conducted an exhaustive systematic review highlighting the Neuroendocrine tumors (NET) patients who reported the effect of treatments on HRQoL. Out of 85 articles, the majority (69) of them are older than five years. One article demonstrated significantly improved HRQoL, while the majority of them did not show any negative impact on HRQoL. The current study proposes sharing of data with the patients in decision making.

One of the most commonly used questionnaires is the European Organization for Research and Treatment of Cancer Quality of Life Questionnaire C30 (EORTC QLQ-C30) which covers physical, role, emotional, social, cognitive along with specific symptoms: pain, fatigue, and nausea and vomiting, dyspnoea, insomnia, appetite loss, constipation, diarrhea, and financial difficulties. The limitation of assessing HRQoL is relying on NET patients to remember and recall the information during their visits rather than real-time data tracking through apps, which may have demonstrated the actual impact of treatment modalities on patients and confidence in the reported data settings of the studies.

Author Response

Thank you very much for your valuable comments.

Reviewer 3 Report

This paper summarizes the data on quality of life of patients with NENs receiving different currently approved therapies. As far as I know it is unique in its subject and highlights an important aspect of decision making not previously elucidated. The authors' proposals on conducting wide-range real-life, cross-sectional, observational or even prospective studies on the quality of life of these patients using standardized and NET-specific questionnaires should be taken into serious consideration by NEN scientific societies . My only suggestion to the authors would be to optimize the included table appearance for better reading. 

Author Response

Thank you very much for your valuable comments. We have made changes to table 1. We have included only RCTs and phase II clinical trials in the main table. We have created another table with the prospective and observational studies and included that in the supplementary file. This limited table 1 to 1.5 pages. 

Reviewer 4 Report

Gosain and colleagues review the impact of different treatment modalities on Health Related Quality of Life (HRQoL) in neuroendocrine tumors.

The manuscript is well written and tables provide a way to have a quick look on manuscript content. In this context, I suggest to authors to add a new table in which both expected and novel symptoms, appering during the pharmachological treatment, are listed.

Author Response

Thank you for your valuable comments. As suggested, we have included another table with the common symptoms that affect the quality of life of patients with NET. We have not included any specific treatment-related symptoms/signs due to specific drugs as the focus of our study was mainly on the symptoms affecting HRQoL.